# Computer cyberspace security mechanism supported by cloud computing

**ZeYuan Fu** [ID] *

School of Cyber Science and Engineering, Wuhan University, Wuhan, China

* 2019302180015@whu.edu.cn

## Abstract

To improve the cybersecurity of Cloud Computing (CC) system. This paper proposes a Network Anomaly Detection (NAD) model based on the Fuzzy-C-Means (FCM) clustering algorithm. Secondly, the Cybersecurity Assessment Model (CAM) based on Grey Relational Grade (GRG) is creatively constructed. Finally, combined with Rivest Shamir Adleman (RSA) algorithm, this work proposes a CC network-oriented data encryption technology, selects different data sets for different models, and tests each model through design experiments. The results show that the average Correct Detection Rate (CDR) of the NAD model for different types of abnormal data is 93.33%. The average False Positive Rate (FPR) and the average Unreported Rate (UR) are 6.65% and 16.27%, respectively. Thus, the NAD model can ensure a high detection accuracy in the case of sufficient data. Meanwhile, the cybersecurity situation prediction by the CAM is in good agreement with the actual situation. The error between the average value of cybersecurity situation prediction and the actual value is only 0.82%, and the prediction accuracy is high. The RSA algorithm can control the average encryption time for very large text, about 12s. The decryption time is slightly longer but within a reasonable range. For different-size text, the encryption time is maintained within 0.5s. This work aims to provide important technical support for anomaly detection, overall security situation analysis, and data transmission security protection of CC systems to improve their cybersecurity.

**Citation:** Fu Z (2022) Computer cyberspace security mechanism supported by cloud computing. PLoS ONE 17(10): e0271546. https://doi.org/10.1371/journal.pone.0271546

**Data Availability Statement:** All relevant data are within the paper and its Supporting Information files.

**Funding:** The author received no specific funding for this work.

## Introduction

The purpose is to improve the security of Cloud Computing (CC). CC is the most popular research direction in the computer field [1]. It has received special attention as new network architecture and network computing pattern. There is an increasing need to improve CC service technology, implement multi-tenant technology, and develop customization functions [2]. For example, virtualization technology can deploy virtual machines on physical ones. It provides users with a virtualized application environment to meet various performance requirements and customize deployment [3, 4]. Generally speaking, the main means of CC is to integrate highly virtual resources on the network and provide services as a service center [5]. Network users can query their own resources on the public server on demand to obtain convenient and fast services [6]. This is the meaning of on-demand resource allocation in CC.

**Competing interests:** The authors have declared that no competing interests exist.

The Internet is the carrier for CC to realize resource sharing. However, the Internet is a heterogeneous and open platform [7] where CC tasks are under security risks. Cyber-attacks include information tampering, intercepting, or deleting, among others. Cybersecurity problems are urgent to promote CC development faster and better. Thus, cloud security is becoming increasingly prominent, showing a trend of diversification and complexity. Meanwhile, it is also imperative to choose proper security technology to protect the environment of tenants [8]. The cloud security alliance research shows that many hackers have attacked the Internet-based CC server. Hackers can use password cracking, secret accounts, dynamic attacks, rainbow tables, botnets, malicious code, and other means [9]. Thus, the CC system must be deployed with safe and effective network protection and monitoring mechanisms.

Based on the above problems, this work first proposes a Network Anomaly Detection (NAD) model based on a clustering algorithm and innovatively uses Fuzzy-C-Means (FCM) clustering algorithm to realize the classification and detection of abnormal data. Secondly, the Cybersecurity Assessment Model (CAM) based on Grey Relational Grade (GRG) is constructed. The Grey Relational Analysis (GRA) method assesses the overall security situation when the network is under attack. Finally, the widely used data encryption algorithm: Rivest Shamir Adleman (RSA) is applied to the CC system. A CC network-oriented data encryption technology based on the RSA algorithm is proposed. At the same time, different data sets are selected for different models, and experiments are designed to test each model. CC is the hot spot of computer development at present. So far, no attacks specifically targeted CC hosts have been found. Nevertheless, a series of virtualization attack threats have emerged in the CC network, such as virtual machine escape, system setup problems, and management program problems. The CC network connects many computer hosts, and these hosts are installed with the same operating system. Thus, Software as a Service (SaaS), Platform as a Service (PaaS), Infrastructure as a Service (IaaS), private cloud, public cloud, and hybrid cloud should be considered in analyzing the problems in the CC host layer. Once a complete problem occurs, these security risks will be quickly spread due to the strong elasticity of CC. This work aims to provide important technical support for anomaly detection, overall security situation analysis, and data transmission security protection of the CC system. Ultimately, it intends to improve CC cybersecurity. The security research under CC has important practical significance for the all-around development of computer networks.

## Material and methods

### NAD model based on clustering algorithm

**(1) Overview of clustering algorithm.** CC systems can facilitate network users by allowing them to share massive amounts of data and computing resources [10]. However, an open mechanism also brings security risks because of algorithm vulnerabilities [11]. In general, there is a certain correlation between similar data. Anomaly detection is a Data Mining process [12] that mines effective, novel, and useful information from big data and discovers underlying relationships and rules in the database [13]. Fig 1 shows ten classic Data Mining algorithms.

The clustering process divides the data in the same dataset into multiple categories (or clusters) under certain standards [14]. The data have a great similarity within a cluster and have great differences between different clusters [15]. K-Means Clustering (KCM) algorithm mainly evaluates the similarity between two samples by calculating their Euclidean Distance (ED) [16]. Fig 2 is the flow of the KMC algorithm.

From the sample space of the dataset, k points are randomly selected as the cluster center. The EC is calculated from each data point to the cluster center. According to the calculation results, the data points are assigned to the cluster closest to the cluster center. Then, the

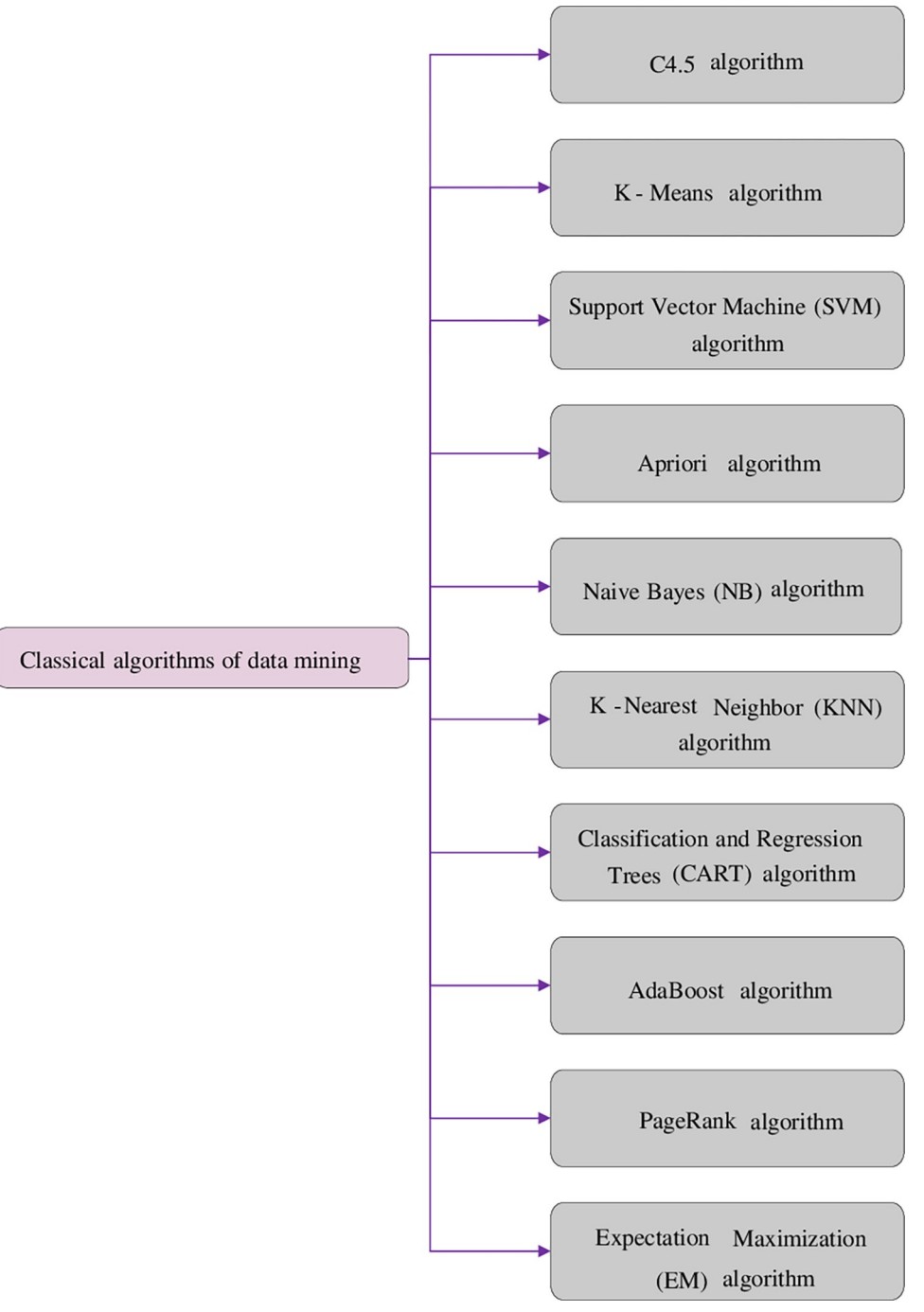

**Fig 1. Classical data mining algorithms.**

iteration begins until each data point has been divided into corresponding clusters. Finally, the data points in each cluster are averaged as the new cluster center. The center is updated step by step until the optimal clustering result is obtained [17].

KMC algorithm strongly depends on the initial k value and the initial cluster center. Moreover, when the amount of data is large, the algorithm needs more iterations to update the cluster center, greatly increasing calculation and memory overhead [18]. Fuzzy C-Means (FCM)

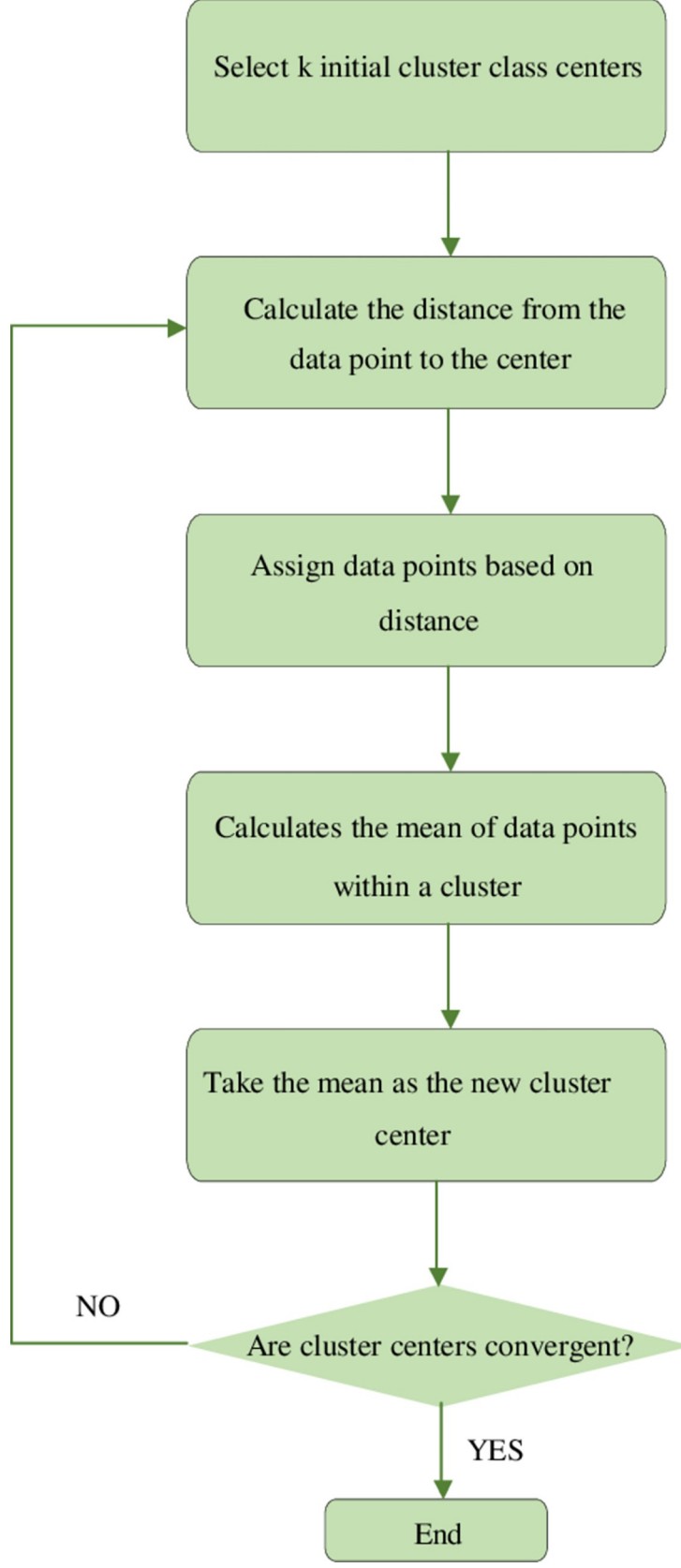

**Fig 2. Flowchart of KMC algorithm.**

clustering improves the KMC in data partition rules. The traditional KMC strictly divides the data points into a cluster and has a high error rate. By comparison, the FCM algorithm adopts fuzzy rules, more flexible in allocating data points. The principle of FCM is to mark each cluster and its members with a Membership Function (MF) and then classify the samples according to the Membership Degree (MD) [19].

**(2) Construction of NAD MODEL.** If the MD of element $X_i$ belonging to category $K$ is $\mu_k$, $k \in [0, 1]$, the MD of $X_i$ belonging to different categories are compared to determine the final category $K$. The sum of the MD of $X_i$ belonging to $K$ categories is 1, that is:

$$\mu_1 + \mu_2 + \cdots \mu_k = 1 \tag{1}$$

FCM clustering algorithm mainly takes the minimum square sum in the cluster as the judgment standard. Suppose the $i$-th cluster center is set as $c_i$, and the weighting index is $m$. In that case, the clustering Objective Function (OF) of dataset $m$ is expressed as:

$$J_m = \sum_{j=1}^{N} \sum_{i=1}^{c} [u_i(X_j)]^m \sqrt{(X_j - c_i)^2} \tag{2}$$

$N$ represents the number of samples; $u_i$ stands for cluster center.

First, the standard format of network data flow includes four basic attributes: source Internet protocol (IP) address, destination IP address, source port, and destination port. Before clustering, the matching degree of the above four attributes of the data is judged according to the matching operation, and the weight of each attribute is preset. Then, the weighting index $m$ is set according to the matching quantity, and $m$ is the sum of the weights of the matching items. The OF expresses the dispersion degree of clustering results. The smaller OF is, the better the clustering effect is.

When OF is the smallest, the FCM clustering algorithm stops iteration. At this time, the partial derivative of the clustering membership of the OF $J_m$ to the $i$-th sample is calculated. The calculation of MF reads:

$$u_i\left(x_j\right) = \frac{[1/\sqrt{u_k(x_{jk} - c_{ik})^2}]^{1/(m-1)}}{\sum_{j=1}^{N} \sum_{i=1}^{c} [1/\sqrt{u_k(x_{jk} - c_{ik})^2}]^{1/(m-1)}} \tag{3}$$

The partial derivative of the OF $J_m$ to the clustering center of the $i$-th sample category $K$ is calculated. The clustering center function is obtained as follows:

$$C_i = \sum_{k=1}^{K} c_{ik}/4 = \sum_{k=1}^{K} \frac{\sum_{j=1}^{N} \sum_{i=1}^{c} [u_i(x_j)]^m x_j}{\sum_{j=1}^{N} \sum_{i=1}^{c} [u_i(x_j)]^m} \tag{4}$$

The four basic attributes of network traffic are abstractly expressed as:

$$F = \{SIP\,(N), DIP\,(N), DP\,(N), SP\,(N)\} \tag{5}$$

$SIP$ represents the weight of the source IP address. $DIP$ is the destination IP address weight. $SP$ denotes the source port weight. $DP$ means the weight of the destination port. Against different types of cyberattacks, the relevant attribute will be weighted differently. Fig 3 gives the workflow of the NAD model based on the FCM clustering algorithm.

In Fig 3, a hierarchical clustering tree is first established with training samples, including a root node and k subtrees. The clustering center is assigned to the subtree nodes, and then the MF is calculated. Next, the correlation analysis is performed on the four basic data flow attributes. The weighted index is calculated as $m$ according to the correlation. Then, the cluster

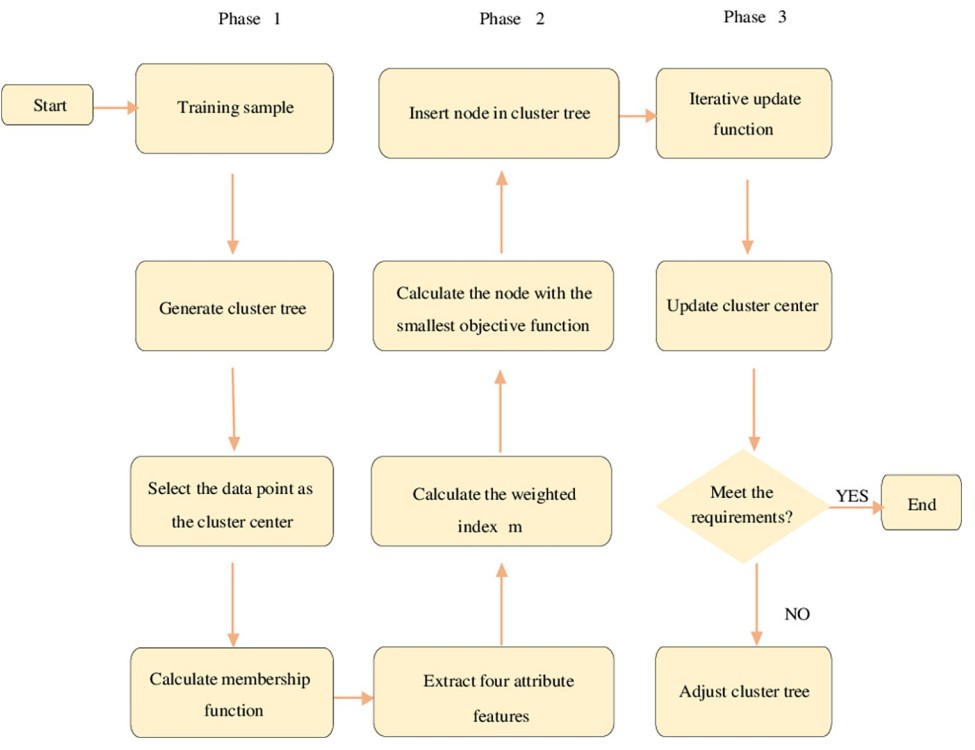

**Fig 3. Workflow of NAD model based on FCM clustering algorithm.**

node with the smallest OF is calculated, and the new data are inserted into the cluster tree node. Finally, the MF and clustering center are updated to determine whether the clustering center has changed. If there is no change, the clustering ends. If there is a change, the iteration continues until the end requirements are met. Here, the communication model includes a physical layer, data link layer, network layer, transport layer, session layer, and application layer.

## Cyberattack Assessment Model (CAM) based on Grey Relational Grade (GRG)

**(1) Grey Relational Analysis (GRA).** Grey theory system has uncertainty. That is, some system information is known, and some are unknown. The research direction is to mine useful information from known information to describe systems with incomplete information. By analyzing the data at a certain level of the system, the grey theory can understand the changes of the system at a higher level, and predict, control, and manage the system. The grey system theory has been widely used because it has no special restrictions on the input data [20]. In simple terms, it analyzes the lower-level data to understand higher-level system changes to evaluate and manage the system [21].

GRA, a branch of grey theory, uses the GRG to express various factors' relationship size, order, and strength [22]. GRG represents the similarity between various factors: the greater the GRG is, the higher the similarity is. Surely, a cyberattack will tamper with the basic attribute of the normal network data. For example, the size of the relationship determines the Cyber Security Index (CSI); that is, the greater the CSI is, the greater the attack will harm the

system [23]. The calculation of GRC-based CSI reads:

$$F_A(t) = 1 - MAX\left(\gamma_1(X, X_1), \gamma_2(X, X_2), \cdots, \gamma_n(X, X_m)\right) \tag{6}$$

$F_A(t)$ represents the CSI attacked by $A$ $t$ times. $X$ is the characteristic sequence of attack data. $X_1, X_2, X_3, \cdots X_m$ denotes the normal network data feature sequence.

**(2) Construction of GRG-based CAM.** Based on the GRA, a CAM is designed, as detailed in Fig 4.

*1) Calculation of service layer CSI.* The calculation of CSI of service layer in time period $t$ reads:

$$F_{S_j}(t) = \sum\nolimits_{i=1}^{n} 10^{P_{ij}} F_{A_j}(t) \tag{7}$$

$A_j$ stands for the cyber-attack. $S_j$ represents service. $F_{A_j}$ denotes the attack situation index generated by $A_j$ against $S_j$. $n$ is the number of attack types $S_j$ received in time $t$. $P_{ij}$ means the harmful degree of $A_j$ to $S_j$, and its value is determined by the type of $A_j$.

*2) Calculation of host layer CSI.* The calculation of CSI of host layer in time period $t$ reads:

$$F_{H_j}(t) = \sum\nolimits_{j=1}^{m} V_j' F_{S_j}(t) \tag{8}$$

$H_j$ indicates the host; $m$ represents the number of services provided by $H_j$; $V_j'$ means the importance of service $S_j$ in all the services provided by $H_j$.

*3) Calculation of system layer CSI.* The calculation of CSI of system layer in time period $t$ reads:

$$F_L(t) = \sum\nolimits_{l=1}^{n} W_j' F_{H_j}(t) \tag{9}$$

$n$ represents the number of hosts in the system; $W_j'$ is the importance of the host $H_j$ in the system.

## CC network-oriented RSA encryption algorithm

**(1) Principle of network data encryption.** Data encryption mainly refers to using a particular encryption algorithm to convert plaintext into ciphertext that cannot be read by non-professionals [24]. On the contrary, converting ciphertext into plaintext is the decryption process and involves certain decryption algorithms [25]. Fig 5 shows the principle of network data encryption.

The sender encrypts the plaintext information into ciphertext and sends it to the recipient. The recipient restores the ciphertext to the original plaintext with the corresponding key. The ciphertext cannot be converted into meaningful information without the corresponding key. Thus, network data are secured [26]. Data encryption can be realized through symmetric or asymmetric encryptions [27]. Fig 6 describes their principle.

Fig 6 implies that symmetric encryption uses the same key for encryption and decryption, whereas asymmetric encryption uses different keys. At present, the most widely used encryption algorithm is the RSA algorithm, asymmetric encryption using the number theory to construct an asymmetric key [28]. The unique encryption mechanism of the RSA makes it almost impossible to be deciphered by a third party and, thus, has high security. Applying the RSA algorithm to a CC system improves network data security substantially [29].

**(2)RSA encryption algorithm.** RSA is an asymmetric encryption algorithm. Thus, RSA must generate a public key for encryption and a private key for decryption. The private key is

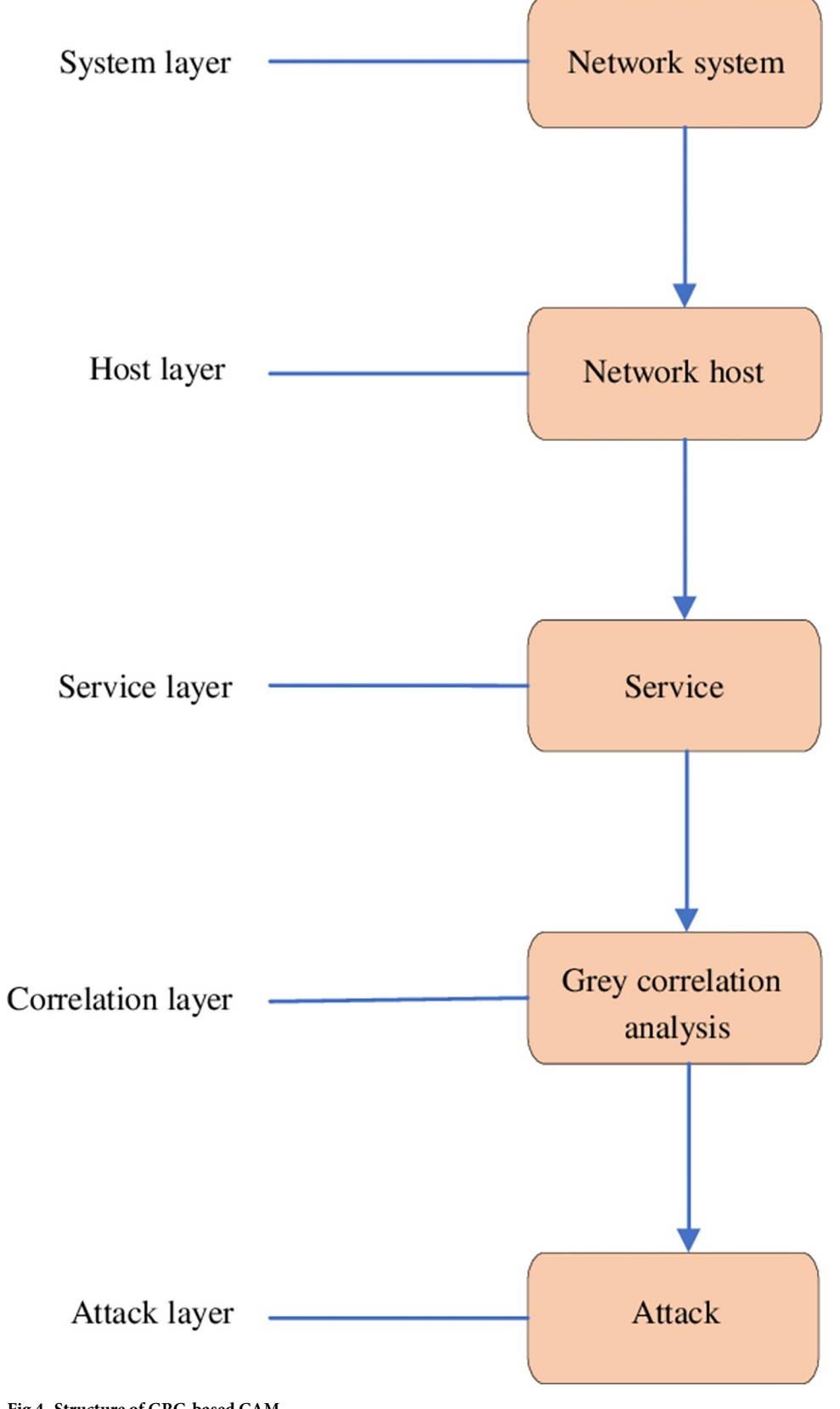

**Fig 4. Structure of GRG-based CAM.**

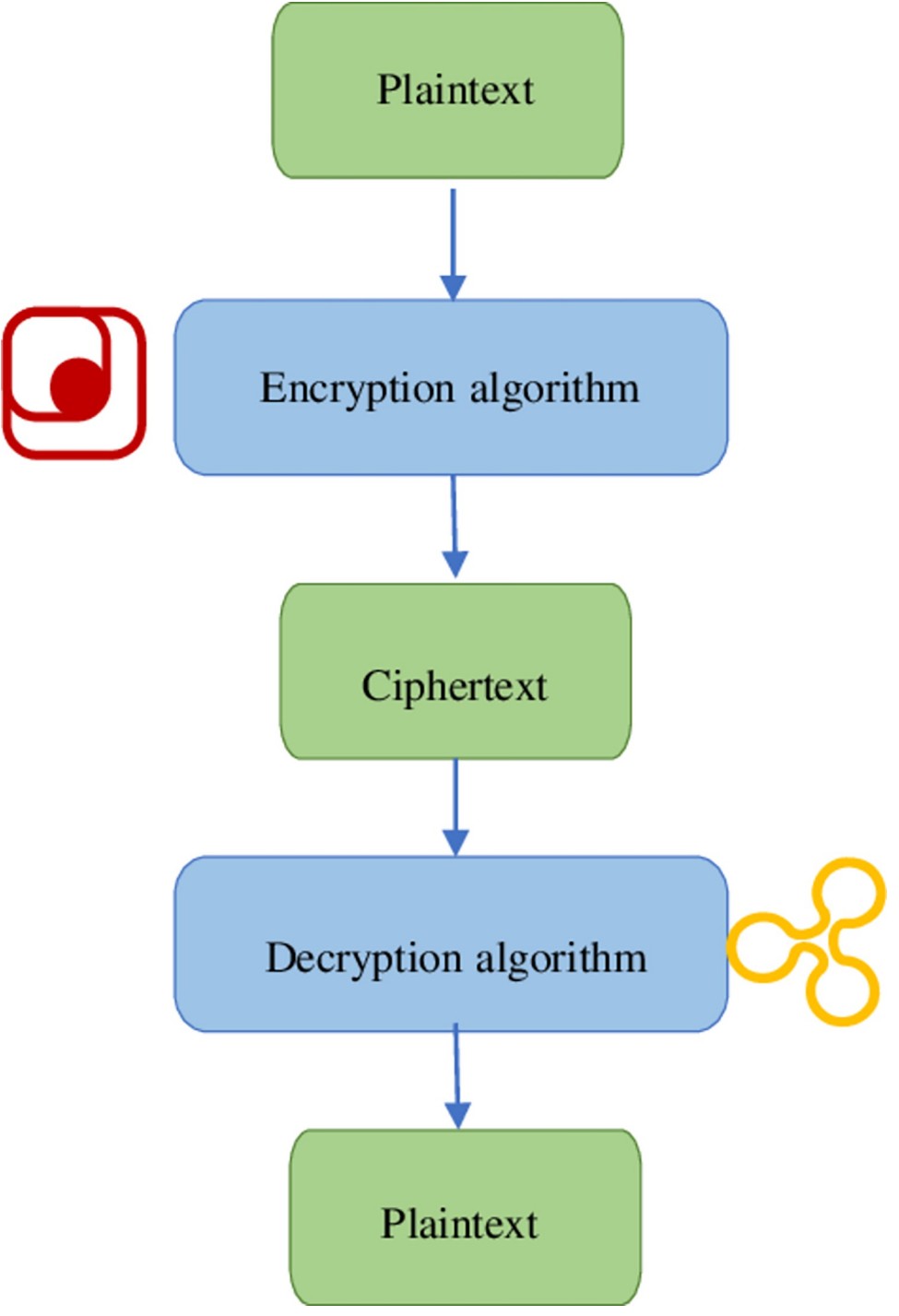

**Fig 5. Schematic diagram of network data encryption.**

only known to the data publisher and receiver to safeguard data transmission. Fig 7 explains the flow of the RSA algorithm.

Step 1: two different prime numbers $p$ and $q$ with long enough digits are selected, and their product is calculated;

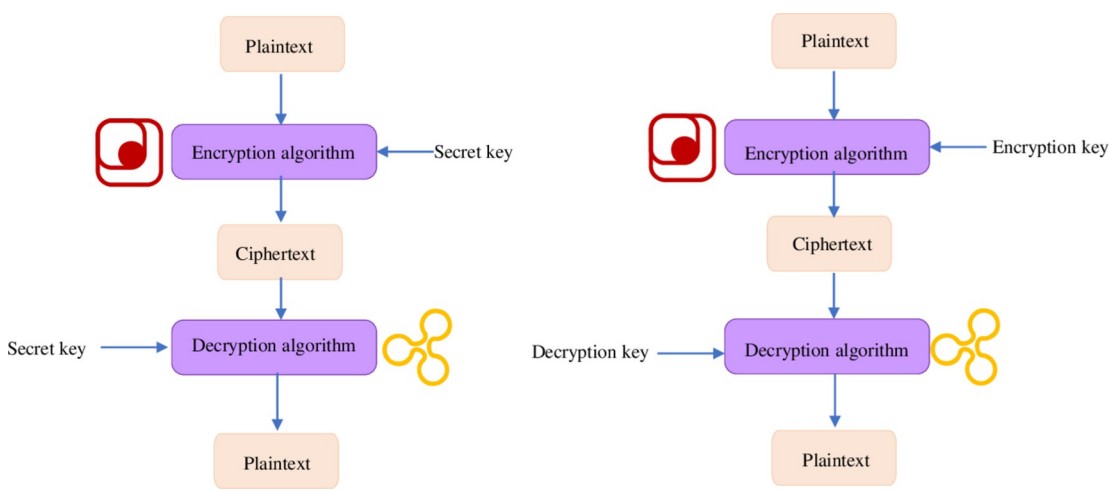

**Fig 6.** Data encryption technology (a. symmetric encryption; b. asymmetric encryption).

Step 2: $f(n) = (p−1)^*(q−1)$ is calculated;

Step 3: an integer $e$, $e>1$ is specified. Moreover, it needs to be less than $f(n)$ and mutual prime;

Step 4: if $e$ and $f(n)$ are known, $d$ is obtained through equation $d^*e≡1modf(n)$, and $mod$ represents the remainder operation;

Step 5: the calculation of encryption (C) and decryption (M) read:

$$C ≡ M^e modn, \ M ≡ C^d modn \qquad (10)$$

## Experimental design

**(1) Testing of FCM-based NAD model.** The knowledge Discovery and Data Mining (KDD) Cup dataset provided by Lincoln Laboratory of Massachusetts Institute of Technology (MIT) is selected to test the performance of the proposed NAD model. In KDD Cup, various user types, different network traffic, and attack means are simulated. It is a dataset dedicated to network anomaly detection. This work randomly selects three data types, including Normal, Denial-Of-Service (DOS), and Probe, from the KDD CUP 99 dataset to construct the model training set. The test set is also randomly selected on the KDD CUP 99 dataset. The evaluation of the proposed FCM model adopts three indexes: False Positive Rate (FPR), Correct Detection Rate (CDR), and Underreported Rate (UR). The data set download website is: http://kdd.ics. uci.edu/databases/kddcup99/kddcup99.html

**(2) Testing GRG-based CAM.** This work selects the Honeynet dataset collected by the famous Honeynet Project to test the proposed GRG-based CAM. The project team attracts

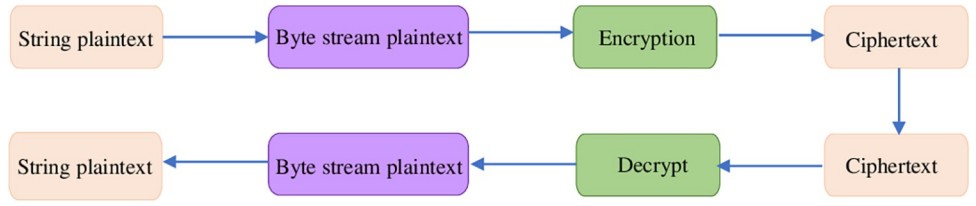

**Fig 7. Flowchart of RSA algorithm.**

various attacks by deliberately setting vulnerabilities in the network and records and analyzing cyberattacks. Overall, attack data over 11 months are collected to construct the Honeynet dataset. Then, it randomly selects data over one month in the dataset and divides them into 1~8 periods. The attack types include Ping, Domain Name System (DNS), and Disk Operating System (DOS).

**(3) Testing of RSA encryption algorithm.** First, the RSA algorithm generates encryptions with different lengths. The test is conducted in ten rounds, and the total and average time costs are recorded. Then, the time cost to encrypt and decrypt a 2M document with a 1,024bit key is recorded. Finally, the time cost of encrypting and decrypting texts of different sizes (1KB, 5KB, 10KB, 20KB, and 40KB) with 1,024-bit and 2,048-bit keys is recorded.

## Results of model test

### Test results of FCM-based NAD model

Fig 8 plots the proposed NAD model's CDR, UR, and FPR on the test dataset.

Fig 8 shows that the CDRs for Normal, DOS, and Probe attacks are 94.27%, 99.93%, and 85.78%, respectively, averaging as high as 93.33%. The URs in the detection process are 5.73%, 0.01%, and 14.22%, respectively, averaging 6.65%. FPRs are 7.35%, 1.91%, and 39.55%, respectively, averaging 16.27%. Obviously, the detection accuracy of the model for the first two types of attack is higher than the Probe attack. Probably, it is due to the different sizes of the data volume. The data volume of Normal and DOS is large, and the model can be well trained, so the accuracy is high. By comparison, Probe type data are much fewer than the other two, so the detection accuracy is low. Therefore, with sufficient data, the proposed network NAD model can ensure high detection accuracy and is suitable for researching anomaly detection in CC systems.

### Test results of GRG-based CAM

Fig 9 displays the number of cyberattacks, the number of hosts attacked, and the evaluation results of the proposed GRG-based CAM over eight time periods on the test dataset.

According to Fig 9A and 9B, the system has been under serious cyberattacks from the third period. Many hosts are attacked in the fifth, sixth, and seventh periods. Thereby, the CSI of these three time periods is relatively high. Fig 9C shows that the proposed GRG-based CAM's predicted average CSI (0.491) is consistent with the actual value (0.487), a 0.82% deviation. Thus, the prediction accuracy is high, and the proposed CRG-based CAM model is suitable for the security evaluation of CC systems.

### Test results of RSA encryption algorithm

**(1) Comparison of results of key generation time.** Fig 10 compares the key generation time under different encryption lengths using the RSA algorithm.

According to Fig 10, the RSA algorithm has taken 23.5s to generate a 1024-bit key in ten rounds of tests, averaging 2.35s every round. By comparison, RSA takes 376.9s to generate a 4096-bit key in ten test rounds, averaging 37.69s every round. Thus, the RSA key generation time increases with the length of the key. Meanwhile, a 1024-bit key can hardly be cracked and only needs about 2s to generate by the RSA algorithm. Thus, the RSA algorithm is effective and secure enough for common applications.

**(2) Comparison of encryption and decryption time.** Fig 11 reveals the time required for encrypting 2M plaintext with 1024-bit key, encrypting and decrypting texts of different sizes with 1024-bit key, and encrypting and decrypting texts of various sizes with 2048-bit key.

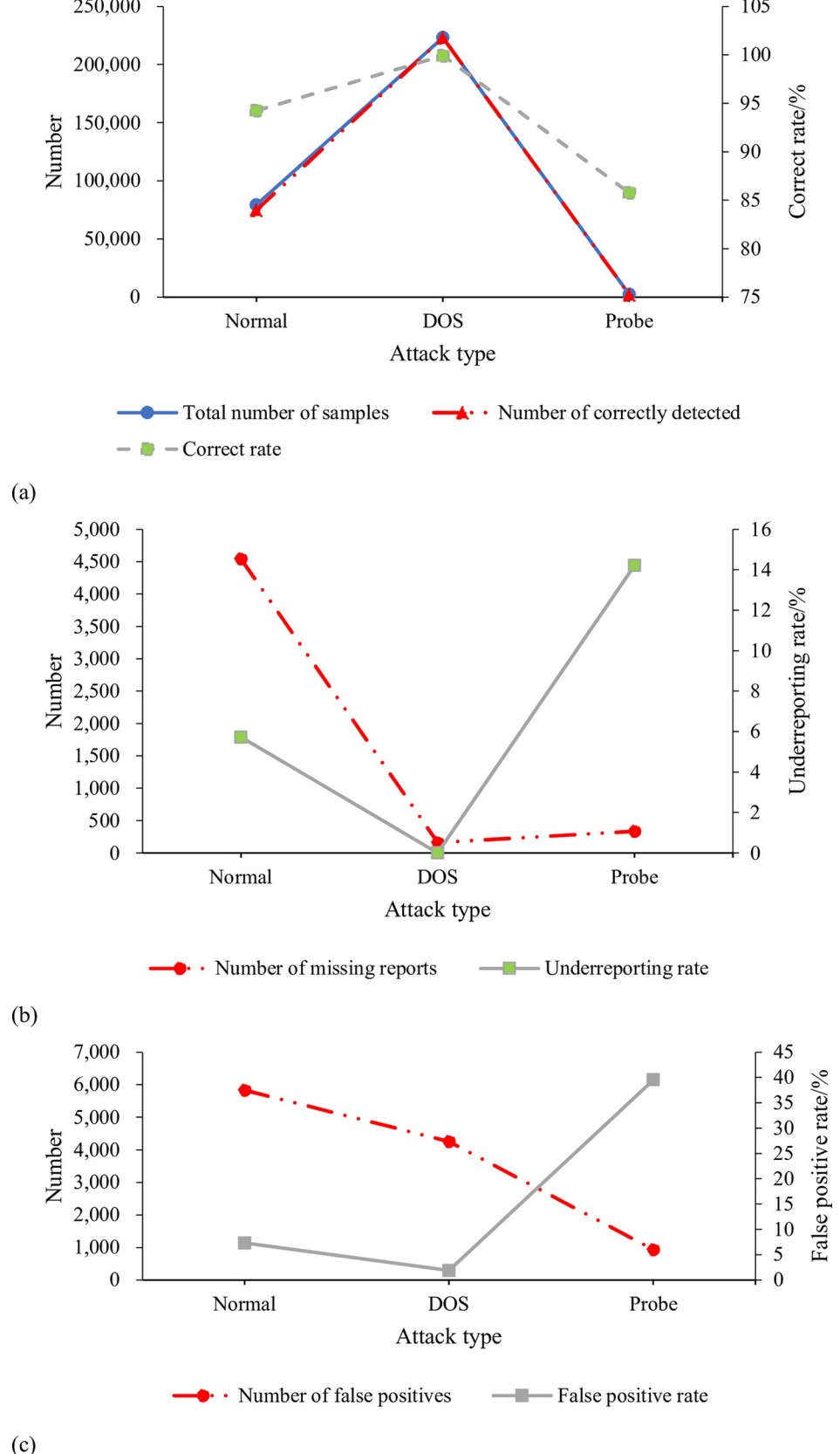

**Fig 8.** Test results of NAD model (a. CDR; b. UR; c. FPR).

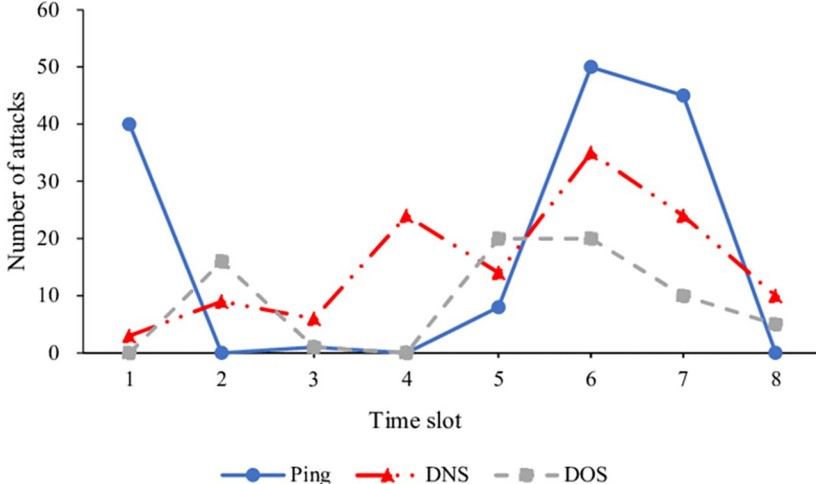

(a)

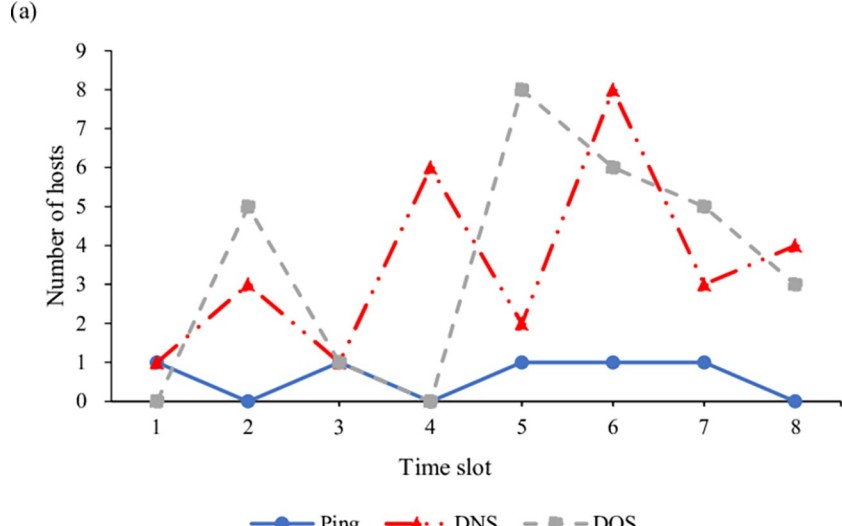

(b)

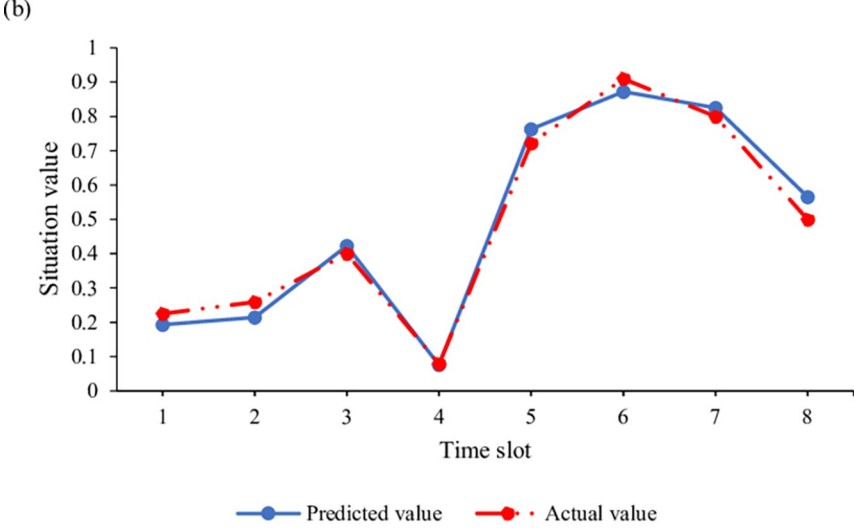

(c)

**Fig 9.** Results of GRG-based CAM (a. number of cyberattacks; b. number of hosts attacked; c. CSI).

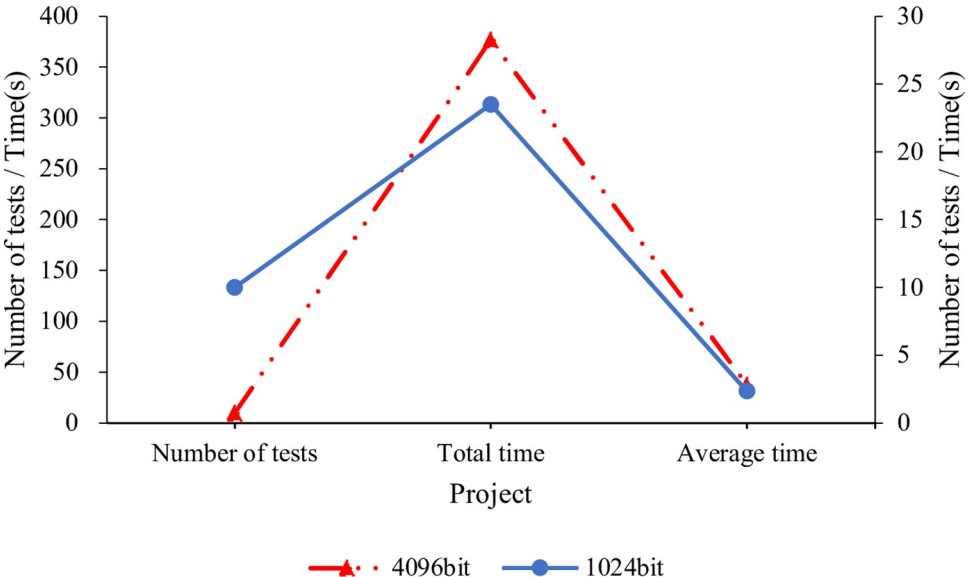

**Fig 10. RSA key generation time.**

Fig 11A suggests that the encryption time and decryption time for 2M text with 1024-bit key is 122.9s and 5611.7s in ten rounds, respectively, averaging 12.29s and 561.17s. Generally, the file reported by the system is less than 100KB, and the average encryption time of the RSA algorithm for 2M large text can also be controlled at about 12s. The decryption time is slightly longer than the encryption time. However, it is also within a reasonable range, so the overall RSA performance is good. Fig 11B and 11C imply that the RSA encrypts length-varying texts smaller than 40KB within 0.3s~0.4s using the 1024-bit key and within 0.5s using the 2048-bit key. Therefore, given a text size of 40KB or less, the encryption time difference between a shorter key and a longer key is small. On the other hand, the decryption time is more than three times that of encryption under a length-constant key. The larger the text is, the longer the decryption time is. The maximum decryption time of a 1024-bit and 2048-bit private key is 9.34s and 31.3s, respectively. In practice, the key length is less than 1024 bits, so the encryption and decryption time will only be much less than the experimental results. Therefore, the RSA algorithm can fully meet the CC system's daily network data encryption requirements.

## Conclusions

The maintenance and stable development of the CC system need to take certain technical measures to discover the cybersecurity risks timely. A NAD model based on a clustering algorithm is proposed here, and a CAM based on GRG is constructed. Finally, it is found that the average CDR of the NAD model for different types of abnormal data is 93.33%, the average UR is 6.65%, and the prediction accuracy is high. The deficiency is that only the widely used and mature algorithms are selected to build the model. Their security performance is not compared with other methods. Therefore, the result is relatively biased, and the threat model is not studied. The follow-up research will select more methods to compare and analyze the security of the proposed model and introduce the threat model to improve the model continuously. The purpose is to provide important technical support for anomaly detection, security situation analysis, data transmission protection of CC system, and improve its cybersecurity. The finding provides a reference for the development of CC security.

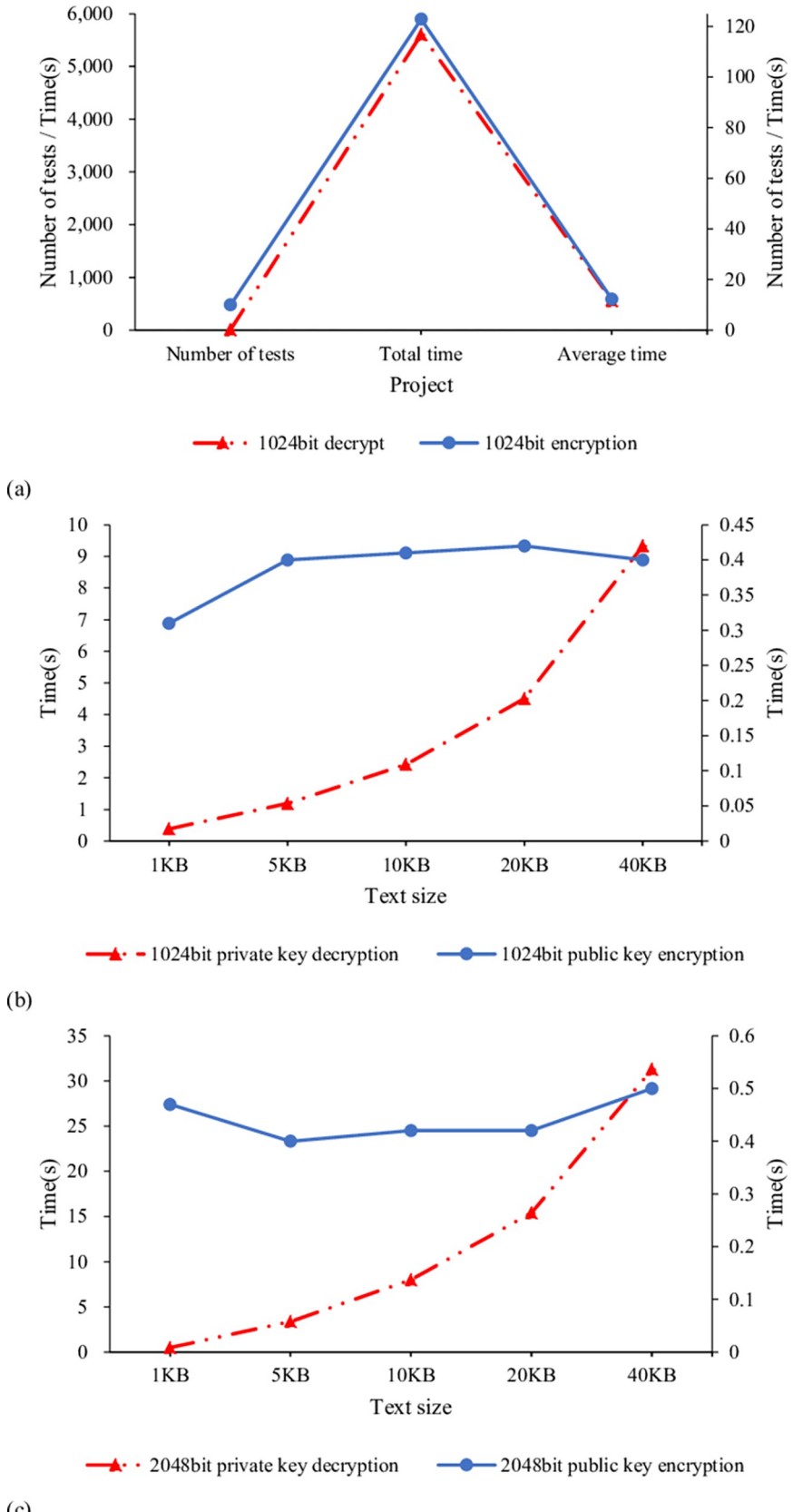

(a)

(b)

(c)

**Fig 11.** Comparison of encryption and decryption time (a. under fixed text size and key length; b. under different sizes of text with 1024-bit key; c. under different sizes of text with 2048-bit key).

## Supporting information

**S1 Data.**
(XLSX)

## Author Contributions

**Conceptualization:** ZeYuan Fu.

**Data curation:** ZeYuan Fu.

**Formal analysis:** ZeYuan Fu.

**Funding acquisition:** ZeYuan Fu.

**Investigation:** ZeYuan Fu.

**Methodology:** ZeYuan Fu.

**Project administration:** ZeYuan Fu.

**Resources:** ZeYuan Fu.

**Software:** ZeYuan Fu.

**Supervision:** ZeYuan Fu.

**Validation:** ZeYuan Fu.

**Visualization:** ZeYuan Fu.

**Writing – original draft:** ZeYuan Fu.

**Writing – review & editing:** ZeYuan Fu.

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
