## [Decision Letter · Decision Letter 0]

24 May 2022

PONE-D-22-07534Computer Cyberspace Security Mechanism Supported by Cloud ComputingPLOS ONE

Dear Dr. Fu,

Thank you for submitting your manuscript to PLOS ONE. After careful consideration, we feel that it has merit but does not fully meet PLOS ONE’s publication criteria as it currently stands. Therefore, we invite you to submit a revised version of the manuscript that addresses the points raised during the review process.

ACADEMIC EDITOR: Please insert comments here and delete this placeholder text when finished. Be sure to:

Please ignore reviewer 3 comments which is irrelevant to this work. Other reviewers feel that the paper lacks novelty. Also, strong English proofreading is required. The authors are also advised to include the following relevant papers in the literature survey. <o:p></o:p>

Efficient NTRU lattice-based certificateless signature scheme for medical cyber-physical systems<o:p></o:p>

A novel proxy-oriented public auditing scheme for cloud-based medical cyber physical systems<o:p></o:p>

Edge-assisted Intelligent Device Authentication in Cyber-Physical Systems<o:p></o:p>

Key management and key distribution for secure group communication in mobile and cloud network<o:p></o:p>

We look forward to receiving your revised manuscript.

Kind regards,

Pandi Vijayakumar, Ph.D

Academic Editor

PLOS ONE

Journal Requirements:

Reviewers' comments:

Reviewer's Responses to Questions

**Comments to the Author**

1. Is the manuscript technically sound, and do the data support the conclusions?

Reviewer #1: Yes

Reviewer #2: No

Reviewer #3: Yes

2. Has the statistical analysis been performed appropriately and rigorously? 

Reviewer #1: Yes

Reviewer #2: No

Reviewer #3: Yes

3. Have the authors made all data underlying the findings in their manuscript fully available?

Reviewer #1: Yes

Reviewer #2: No

Reviewer #3: Yes

4. Is the manuscript presented in an intelligible fashion and written in standard English?

Reviewer #1: Yes

Reviewer #2: No

Reviewer #3: Yes

5. Review Comments to the Author

Reviewer #1: Authors presented Computer Cyberspace Security Mechanism Supported by Cloud Computing in this paper. This paper has merit and covered an important topic, however, I have following suggestions to improve the quality of this paper:

-Explain your contribution in better way.

-Why this kind of study on Computer Cyberspace Security Mechanism Supported by Cloud Computing is important?

- Paper needs to polish and provide a detailed explication of theoretical aspects such as conditions and theorems, and practical issues like algorithms, rules and possible applications.

-Improve the quality of figures.

-The abstract, Introduction and conclusion sections are poor and need to be rewritten to point out significance and impact of the paper.

-I will encourage the authors to spend more time to perform and add some more experiments in the results section.

-remove all typos and other grammatical errors.

-Explain novelty of your work presented in this work.

-Remove all the typos.

-The authors are advised to refer some more recent, relevant and high quality research works. For example:

Blockchain-assisted secure fine-grained searchable encryption for a cloud-based healthcare cyber-physical system,

A reputation score policy and Bayesian game theory based incentivized mechanism for DDoS attacks mitigation and cyber defense,

Secure and energy efficient-based E-health care framework for green internet of things,

A trust infrastructure based authentication method for clustered vehicular ad hoc networks,

IoT transaction processing through cooperative concurrency control on fog–cloud computing environment

The formula character format is best to be different from the main text, and mathematical characters are recommended.

Also, some security related researches may also be explored and discussed:

Defending deep learning models against adversarial attacks,

Secure blockchain enabled Cyber-physical systems in healthcare using deep belief network with ResNet model,

Defense mechanisms against DDoS attack based on entropy in SDN-cloud using POX controller,

Many references are with incomplete bibliographic information (like lack of publication venue, for instance). This must be corrected

The formula character format is best to be different from the main text, and mathematical characters are recommended.

It seems that the contribution points of the article are a little bit few. After or in the section of Motivation, it is recommended that the authors summarize the contribution points of their work, which clearly demonstrate the innovations.

Moreover, the format of the references should strictly follow the rules of the journal.

Reviewer #2: Authors discussed about computer cyberspace security mechanism Supported by Cloud Computing.

Paper seems very weak in its current form. It should be revised as per the following comments:

* Add communication model of the considered communication environment in the paper.

* Add threat model in the paper.

* Add comparative performance analysis of the various security protocols of this domain.

* Improve the English writing of the paper.

* Highlight the research contributions of the paper.

* What is the motivation of the conducted study.

Reviewer #3: The authors addressed all my review comments satisfactoryly. Now this paper looks good in technological aspects. Hence I strongly recommend this paper for possible publication in your reputed journal.

6. PLOS authors have the option to publish the peer review history of their article (what does this mean?). If published, this will include your full peer review and any attached files.

Reviewer #1: No

Reviewer #2: No

Reviewer #3: No

---

## [Author Response · Author response to Decision Letter 0]

16 Jun 2022

Date: May 24 2022 06:33AM

To: "Zeyuan Fu" 2019302180015@whu.edu.cn

From: "PLOS ONE" plosone@plos.org

Subject: PLOS ONE Decision: Revision required [PONE-D-22-07534]

PONE-D-22-07534

Computer Cyberspace Security Mechanism Supported by Cloud Computing

PLOS ONE

Dear Dr. Fu,

Thank you for submitting your manuscript to PLOS ONE. After careful consideration, we feel that it has merit but does not fully meet PLOS ONE’s publication criteria as it currently stands. Therefore, we invite you to submit a revised version of the manuscript that addresses the points raised during the review process.

Reply: Thank you for your review.

ACADEMIC EDITOR: Please insert comments here and delete this placeholder text when finished. Be sure to:

Please ignore reviewer 3 comments which is irrelevant to this work. Other reviewers feel that the paper lacks novelty. Also, strong English proofreading is required. The authors are also advised to include the following relevant papers in the literature survey. 

Efficient NTRU lattice-based certificateless signature scheme for medical cyber-physical systems

A novel proxy-oriented public auditing scheme for cloud-based medical cyber physical systems

Edge-assisted Intelligent Device Authentication in Cyber-Physical Systems

Key management and key distribution for secure group communication in mobile and cloud network

Reply: Thank you for your advice. We have ignored reviewer 3's comments. In the introduction part, the author adds some innovative points and proofreads the English writing of the article. The four literatures you mentioned are cited.

A rebuttal letter that responds to each point raised by the academic editor and reviewer(s). You should upload this letter as a separate file labeled 'Response to Reviewers'.

A marked-up copy of your manuscript that highlights changes made to the original version. You should upload this as a separate file labeled 'Revised Manuscript with Track Changes'.

An unmarked version of your revised paper without tracked changes. You should upload this as a separate file labeled 'Manuscript'.

Reply: Thank you for your comments. We will upload the revised version of the article according to your requirements.

We look forward to receiving your revised manuscript.

Kind regards,

Pandi Vijayakumar, Ph.D

Academic Editor

PLOS ONE

Reply: Thank you again for your review. 

Journal Requirements:

Reply: Thank you for your advice. We have adjusted the article layout according to the format requirements.

Reply: Thank you for your advice. The dataset Download URL has been added to the "experimental design" section. And add the data package in the attachment.

Reply: Thank you for your comments. We have obtained and verified the ORCID.

Reply: Thank you for your reminder. We have adjusted the abstract on the online submission form accordingly.

Reviewers' comments:

Reviewer's Responses to Questions

Comments to the Author

1. Is the manuscript technically sound, and do the data support the conclusions?

Reviewer #1: Yes

Reviewer #2: No

Reviewer #3: Yes

2. Has the statistical analysis been performed appropriately and rigorously?

Reviewer #1: Yes

Reviewer #2: No

Reviewer #3: Yes

3. Have the authors made all data underlying the findings in their manuscript fully available?

Reviewer #1: Yes

Reviewer #2: No

Reviewer #3: Yes

4. Is the manuscript presented in an intelligible fashion and written in standard English?

Reviewer #1: Yes

Reviewer #2: No

Reviewer #3: Yes

Reply: Thank all the reviewers for their comments on this article. We will revise the article according to the specific comments below.

5. Review Comments to the Author

Reviewer #1: Authors presented Computer Cyberspace Security Mechanism Supported by Cloud Computing in this paper. This paper has merit and covered an important topic, however, I have following suggestions to improve the quality of this paper:

-Explain your contribution in better way.

Reply: Thank you for your advice. The research contribution of this paper has been added at the end of the introduction.

-Why this kind of study on Computer Cyberspace Security Mechanism Supported by Cloud Computing is important?

Reply: Explanations have been added at the end of the introduction.

- Paper needs to polish and provide a detailed explication of theoretical aspects such as conditions and theorems, and practical issues like algorithms, rules and possible applications.

Reply: Thank you for your advice. More rationale has been added in the "Cyberattack Assessment Model (CAM) based on Grey Relational Grade (GRG)" section.

-Improve the quality of figures.

Reply: Thank you for your comments. The full text figures have been adjusted.

-The abstract, Introduction and conclusion sections are poor and need to be rewritten to point out significance and impact of the paper.

Reply: The abstract, introduction, and conclusion have been rewritten.

-I will encourage the authors to spend more time to perform and add some more experiments in the results section.

Reply: Thank you for your support. For more experimental results, we will conduct in-depth research in the future, which is also one of the future directions of the conclusion part.

-remove all typos and other grammatical errors.

Reply: Thank you for your reminder. English writing has been re-proofread.

-Explain novelty of your work presented in this work.

Reply: Innovations have been added to the introduction.

-Remove all the typos.

Reply: Thank you for your reminder. English writing has been re-proofread.

-The authors are advised to refer some more recent, relevant and high quality research works. For example:

Blockchain-assisted secure fine-grained searchable encryption for a cloud-based healthcare cyber-physical system,

A reputation score policy and Bayesian game theory based incentivized mechanism for DDoS attacks mitigation and cyber defense,

Secure and energy efficient-based E-health care framework for green internet of things,

A trust infrastructure based authentication method for clustered vehicular ad hoc networks,

IoT transaction processing through cooperative concurrency control on fog–cloud computing environment

Reply: Thank you for your advice. We have cited the references you provided.

The formula character format is best to be different from the main text, and mathematical characters are recommended.

Reply: Thank you for your advice. We have confirmed that all formula characters are mathematical characters.

Also, some security related researches may also be explored and discussed:

Defending deep learning models against adversarial attacks,

Secure blockchain enabled Cyber-physical systems in healthcare using deep belief network with ResNet model,

Defense mechanisms against DDoS attack based on entropy in SDN-cloud using POX controller,

Reply: Thank you for your advice. These references have been added to the list of references.

Many references are with incomplete bibliographic information (like lack of publication venue, for instance). This must be corrected

Reply: Thank you for your comments. We have perfected the references.

The formula character format is best to be different from the main text, and mathematical characters are recommended.

Reply: Thank you for your advice. We have confirmed that all formula characters are mathematical characters.

It seems that the contribution points of the article are a little bit few. After or in the section of Motivation, it is recommended that the authors summarize the contribution points of their work, which clearly demonstrate the innovations.

Reply: Thank you for your advice. Article contribution and innovation have been added in the introduction.

Moreover, the format of the references should strictly follow the rules of the journal.

Reply: We have ensured that the format of the references meets the requirements of the journal.

Reviewer #2: Authors discussed about computer cyberspace security mechanism Supported by Cloud Computing.

Paper seems very weak in its current form. It should be revised as per the following comments:

* Add communication model of the considered communication environment in the paper.

Reply: Thank you for your comments. The communication model has been added at the end of the “Intrusion Detection Model (IDM) based on clustering algorithm” section.

* Add threat model in the paper.

Reply: Thank you for your comment. The construction and research of threat model will be carried out in-depth in future research, which has been added to the conclusion.

* Add comparative performance analysis of the various security protocols of this domain.

Reply: Thank you for your advice. The comparative performance analysis of security protocols already exists in the future direction of the conclusion, and we will conduct in-depth research in this field in the future.

* Improve the English writing of the paper.

Reply: Thank you for your reminder. English writing has been proofread.

* Highlight the research contributions of the paper.

Reply: The contribution of the thesis has been highlighted in the abstract, introduction and conclusion.

* What is the motivation of the conducted study.

Reply: Research motivation has been added in the abstract and introduction.

Reviewer #3: The authors addressed all my review comments satisfactoryly. Now this paper looks good in technological aspects. Hence I strongly recommend this paper for possible publication in your reputed journal.

Reply: Thank you for your comment.

6. PLOS authors have the option to publish the peer review history of their article (what does this mean?). If published, this will include your full peer review and any attached files.

Do you want your identity to be public for this peer review? For information about this choice, including consent withdrawal, please see our Privacy Policy.

Reviewer #1: No

Reviewer #2: No

Reviewer #3: No

Reply: Thank you again for your support for this study.

---

## [Decision Letter · Decision Letter 1]

4 Jul 2022

Computer Cyberspace Security Mechanism Supported by Cloud Computing

PONE-D-22-07534R1

Dear Dr. Fu,

We’re pleased to inform you that your manuscript has been judged scientifically suitable for publication and will be formally accepted for publication once it meets all outstanding technical requirements.

Kind regards,

Pandi Vijayakumar, Ph.D

Academic Editor

PLOS ONE

Additional Editor Comments (optional):

Reviewers' comments:

Reviewer's Responses to Questions

**Comments to the Author**

1. If the authors have adequately addressed your comments raised in a previous round of review and you feel that this manuscript is now acceptable for publication, you may indicate that here to bypass the “Comments to the Author” section, enter your conflict of interest statement in the “Confidential to Editor” section, and submit your "Accept" recommendation.

Reviewer #1: All comments have been addressed

Reviewer #2: All comments have been addressed

2. Is the manuscript technically sound, and do the data support the conclusions?

Reviewer #1: Yes

Reviewer #2: Yes

3. Has the statistical analysis been performed appropriately and rigorously? 

Reviewer #1: Yes

Reviewer #2: Yes

4. Have the authors made all data underlying the findings in their manuscript fully available?

Reviewer #1: Yes

Reviewer #2: Yes

5. Is the manuscript presented in an intelligible fashion and written in standard English?

Reviewer #1: Yes

Reviewer #2: Yes

6. Review Comments to the Author

Reviewer #1: Computer Cyberspace Security Mechanism Supported by Cloud Computing is presented in this paper and it is revised well.

Reviewer #2: Paper has been updated as per the comments provided in the previous round of review. The quality of the paper has been improved. I recommend acceptance of the paper.

7. PLOS authors have the option to publish the peer review history of their article (what does this mean?). If published, this will include your full peer review and any attached files.

Reviewer #1: No

Reviewer #2: No

---

## [Editor Report · Acceptance letter]

19 Sep 2022

PONE-D-22-07534R1 

Computer Cyberspace Security Mechanism Supported by Cloud Computing 

Dear Dr. Fu:

I'm pleased to inform you that your manuscript has been deemed suitable for publication in PLOS ONE. Congratulations! Your manuscript is now with our production department. 

Kind regards, 

on behalf of

Dr. Pandi Vijayakumar 

Academic Editor

PLOS ONE